# FiD-Light: Efficient and Effective Retrieval-Augmented Text Generation

## Abstract

Retrieval-augmented generation models offer many benefits over standalone language models: besides a textual answer to a given query they provide provenance items retrieved from an updateable knowledge base. However, they are also more complex systems and need to handle long inputs. In this work, we introduce FiD-Light to strongly increase the efficiency of the state-of-the-art retrieval-augmented FiD model, while maintaining the same level of effectiveness. Our FiD-Light model constrains the information flow from the encoder (which encodes passages separately) to the decoder (using concatenated encoded representations). Furthermore, we adapt FiD-Light with re-ranking capabilities through textual source pointers, to improve the top-ranked provenance precision. Our experiments on a diverse set of seven knowledge intensive tasks (KILT) show FiD-Light consistently improves the Pareto frontier between query latency and effectiveness. FiD-Light with source pointing sets substantial new state-of-the-art results on six KILT tasks for combined text generation and provenance retrieval evaluation, while maintaining reasonable efficiency.

## 1 Introduction

Enabling machine learning models to access information contained in parametric or non-parametric storage (i.e., retrieval-enhanced machine learning) can lead to efficiency and/or effectiveness improvements in a wide range of learning tasks (Zamani et al., 2022). For example, retrieval-augmented generation (Lewis et al., 2020), which is the focus of this paper, has a manifold of benefits over closed-loop language modelling in knowledge intensive tasks: Answers can be grounded in (multiple) specific pieces of information which enables clear attribution (Dehghani et al., 2019; Rashkin et al., 2021; Lamm et al., 2021); the knowledge base can easily be managed, updated, and swapped (Izacard et al., 2022); the decomposition of retrieval and generation module offers clear efficiency-effectiveness tradeoff controls; and the data structure of combined retrieval and text generation enables many insightful failure analyses. However, with these benefits also come downsides, such as a higher system complexity with higher training and inference cost. Therefore, our goal is to reduce costs as much as possible, while retaining effectiveness, to make these benefits more widely available.

The most effective approach for knowledge intensive tasks, such as those contained in the KILT benchmark (Petroni et al., 2021), is the Fusion-in-Decoder (FiD) model proposed by Izacard & Grave (2020). The FiD model uses an external retriever, such as a dense retrieval model, to gather candidate passages, which are encoded with the query by a T5-encoder (Raffel et al., 2020); the encoded vectors are concatenated and fed through a T5-decoder to produce a single output string. FiD can synthesize answers from multiple different sources, which leads to state-of-the-art results in many tasks from open domain QA to fact verification (Hofstätter et al., 2022; Izacard et al., 2022).

While undoubtedly the leading architecture – in terms of effectiveness for knowledge intensive generation tasks – the FiD model is resource intensive. In state-of-the-art configurations concatenating all encoded tokens before the decoding leads often to sequences longer than 10 thousand vectors, coupled with auto-regressive decoding, this leads to a high inference latency. In Figure 1 we plot the average latency of a single query measured on a single TPUv4 of the encoder and decoder modules

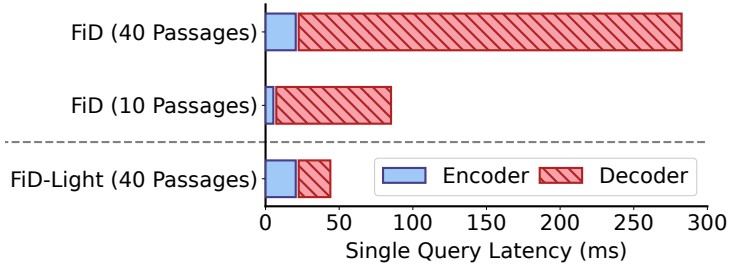

Figure 1: Average inference latency for a query of FiD & FiD-Light (T5-Base on a single TPUv4).

of FiD.[1] The first observation is the overpowering 93% of time spent on decoding in FiD. A common and straightforward approach to reduce the latency of FiD is to reduce the number of input passages, e.g., to only 10 passages. While this approach naturally reduces the overall latency, the decoding latency still requires 10 times as long as the encoding (see Figure 1). Crucially, this approach will also reduce the model's effectiveness substantially, as we show later in this work (see §4.3).

To overcome the inefficiencies of the decoding, we propose *FiD-Light*, a simple yet effective adaptation of the FiD model. The connection between the encoder and decoder has a large capacity for information in FiD. In contrast, the retrieval community, showed that in applications, such as dense retrieval with dot-product scoring, encoded information may be compressed to a fraction of the original input length, including representing passages in a single (Hofstätter et al., 2021) or multiple vectors (Chen et al., 2020). Following these footsteps, we propose to compress the number of vectors per encoded passage, to a fraction of the input vectors, before they are accessed by the decoder. Using this approach FiD-Light is able to ingest a large number of passages with strongly reduced latency, as illustrated in Figure 1. Here we still use 40 passages, showing the same encoding time as FiD, but a substantially faster decoding (now on par with the encoding time), for a total latency lower than FiD with 10 passages.

The knowledge intensive tasks we aim to solve ideally require a system to produce both a generated output text, as well as a ranked list of provenance items from the knowledge base. However, FiD is limited to only produce output text. Falling back to return the original candidate ranking is usually sub-optimal with low-precision. To incorporate re-ranking capabilities into FiD-Light we adapt a passage marker workflow proposed by Lakhotia et al. (2021) as part of FiD-Ex. They marked the input passages with textual indices, and trained the model to output the relevant indices in the output text. We find that using these textual indices or *source pointers* directly as output, as Lakhotia et al. (2021) proposed, is brittle and prone to distribution shifts in the number of expected relevant passages between training and evaluation (see §4.2). Therefore, our *FiD-Light^{SP}* approach re-ranks the selected passages to the top of the ranked list, without discarding the rest of the retrieved list, for higher robustness and improved results.

We conduct experiments on seven tasks of the KILT benchmark composed by Petroni et al. (2021) spanning open domain QA, slot filling, fact verification, and dialogue tasks. We study the following research questions to demonstrate the efficacy of our proposed FiD-Light^{SP} model:

**RQ1** What impact does training the retrieval module have on FiD-Light^{SP} downstream results?

The quality of the final result is strongly bound by the recall quality of the retriever module. While many complex end-to-end training procedures have been proposed (Singh et al., 2021; Izacard et al., 2022), we focus on simple, yet effective directly supervised dense retrieval training. We show that a simple retrieval training comfortably outperforms a zero-shot retrieval baseline from Hofstätter et al. (2022) and the resulting FiD-Light^{SP} downstream results take a major step towards a realistic oracle retriever ceiling.

**RQ2** How robust is our source pointing and re-ranking workflow applied to FiD and FiD-Light?

We use available passage relevance information for each task in the KILT benchmark to train our source pointer output via text markers. We train the FiD(-Light) generator to output the indices for

---

[1]All our measurements in this work are conducted on TPUv4s, however we confirmed that using V100 GPUs we observe a similar ratio of time spent in the encoder vs. the decoder of FiD and FiD-Light.

all relevantly retrieved passages during training, before generating the textual answer. We observe that FiD(-Light)$^{SP}$ is learning an expected distribution for the number of selected passages, which might not match relevance distributions during evaluation. To mitigate this problem we propose to use the source pointer to re-rank the initial list. We show this improves the results over FiD-Ex. Comparing the effectiveness of the source pointers between different FiD-Light settings and the FiD baseline we find FiD$^{SP}$ to rapidly lose effectiveness when the number of input passages is reduced, while FiD-Light$^{SP}$ is able to hold the passage precision at much lower latency.

**RQ3** How does FiD-Light$^{SP}$ compare to the FiD$^{SP}$ baseline in efficiency-effectiveness tradeoffs?

The common approach to speed up FiD is to reduce the number of input passages. To this we compare our FiD-Light$^{SP}$ model using a static number of passages, but varying the number of vectors fed into the decoder as well as changing the T5 backbone size. We show that while FiD$^{SP}$ with fewer passages strongly degrades, FiD-Light$^{SP}$ is able to hold most of the initial maximum effectiveness of FiD$^{SP}$, while being $3\times$ faster. This Pareto optimal result between latency and effectiveness is complemented when we increase the T5-backbone sizes in FiD-Light$^{SP}$ to receive the benefits of larger models, while still outperforming the initial FiD$^{SP}$ baseline in terms of efficiency. Overall FiD-Light$^{SP}$ is Pareto optimal on six out of the seven tested tasks.

**RQ4** How does FiD-Light$^{SP}$ compare to related methods on the KILT benchmark?

We submitted three representative configurations of FiD-Light$^{SP}$ to the blind-evaluated KILT leaderboard test set to compare them to other methods for knowledge intensive tasks. We evaluate FiD-Light$^{SP}$ on the main metric of the KILT benchmark: combined KILT-scores (which only counts a text generation score if the R-Precision for the query is 1). We show FiD-Light$^{SP}$ outperforms previous SOTA models by considerable margins on the KILT-scores on six tasks. We set new SOTA results compared to the previous best methods on:

- **QA** *HotpotQA* +11.1 K-EM (+61.3%), *NQ* +7.5 K-EM (+17.2%), *TriviaQA* +5.8 K-EM (+10.0%)
- **Slot Filling** *zsRE* +10.8 K-AC (+14.8%), *T-REx* +0.5 K-AC (+0.7%)
- **Fact Verification** *FEVER* +6.0 K-AC (+7.6%)

We hope these results demonstrate to the community that SOTA results are achievable with reasonable efficiency and that efficient retrieval-augmented generation has a promising future ahead.

## 2 BACKGROUND AND RELATED WORK

In this section, we first review the FiD model and FiD-Ex workflow, which adds textual explanation markers to FiD. We further discuss other related work in this area.

### 2.1 FID (FUSION IN DECODER) WITH EXPLANATIONS

A critical capability for retrieval-augmented models is to be able to synthesize and utilize information from multiple distinct retrieved items (Zamani et al., 2022). To effectively implement this paradigm Izacard & Grave (2020) proposed the FiD model, which re-wires the computational graph between an of-the-shelf pre-trained Transformer Encoder and Decoder (Vaswani et al., 2017). Usually FiD is initialized with the pre-trained T5 model (Raffel et al., 2020). Given a query $q$, we retrieve a set of $n$ candidate passages using a separate retrieval module. The retriever is independently trained, and can take any traditional, neural or hybrid architecture. As in Izacard & Grave (2020), we use a single dense retriever, as it has been shown to outperform traditional retrieval methods (Hofstätter et al., 2022). To encode the information, FiD concatenates the query $q$ with each retrieved passage $p$ and independently feeds (one per index $i$) the sequences through a Transformer encoder ($T_E$):

$$e_i = \mathrm{T_E}([\text{``query: ''}; q; \text{``context: ''}; p_i])  \qquad (1)$$

The resulting encoded representations – using one vector per token – are concatenated into a single long sequence, which is fed through the Transformer decoder ($T_D$), autoregressively during inference, to produce a single output sequence $o$:

$$o = \mathrm{T_D}([e_1; e_2; ...; e_n])  \qquad (2)$$

FiD has two main limitations: (1) the text-only output does not provide any information about the exact passage(s) which were used to synthesize the output; and (2) the long input sequence leads to highly inefficient autoregressive decoding (as shown in Figure 1). While the expected output is relatively short (in the magnitude of dozens of tokens), the input to the decoder is large with $O(n * (|q| + |p|))$ tokens (in the magnitude of thousands of tokens).

To alleviate limitation (1) Lakhotia et al. (2021) adapt the FiD workflow with textual *explanations* (FiD-Ex) inspired by the WT5 (Why?, T5) concept proposed by Narang et al. (2020). For FiD-Ex, the FiD architecture is left untouched; Lakhotia et al. (2021) only adapt the textual input and target output. The input to the encoder is augmented with indices (from $1$ to $n$) to identifiy individual passages:[2]

$$e_i = \mathrm{T_E}([\text{"query: "}; q; \text{"index: "}; i; \text{"context: "}; p_i]) \tag{3}$$

And the target output $t$ during training is augmented with the indices (using the regular tokens for the numbers and spaces as separators for multiple indices) of all the known relevant passages $R^+$ in the retrieved set:

$$\hat{t} = [\text{"index: "}; \{r | r \in R^+\}; \text{"text: "}; t] \tag{4}$$

On one hand, this textual formulation packs more capabilities in the same text based architecture, on the other hand we note that this discrete selection of the top-$|R^+|$ passages from the candidate set is a strong departure from the prevalent pairwise re-ranking models. It opens a new range of induced biases about expected distributions of $|R^+|$ not studied before. During inference the output is parsed to extract the indices as numbers and remove the additional textual markers to evaluate the output text.

## 2.2 RELATED WORK

**Efficient Generation Models.** To enable their ubiquitous use, a key component besides their safety, is the efficiency of text generators to run at scale. Naturally, many studies work to achieve this goal from various angles. Schuster et al. (2022) propose an adaptive early exiting language model, which exits the decoder stack of Transformer layers early for easy to predict tokens. The LongT5 model focuses on improving the efficiency of the encoder for long input sequences (Guo et al., 2021), in contrast we focus on the decoder efficiency, as FiD's encoder input is usually short. We believe our FiD-Light adaptations are orthogonal to many other algorithmic and engineering-based generation efficiency improvements and can be combined in future work. For a comprehensive overview over efficient transformer architectures, we refer the reader to Tay et al. (2022).

**Retrieval-Enhanced Machine Learning.** The foundational retrieval-augmented models, e.g., FiD (Izacard & Grave, 2020), RAG (Lewis et al., 2020), and REALM, (Guu et al., 2020) are trained to solve individual tasks. Many of their recent improvements optimized end-to-end processes (e.g., EMDR2 (Singh et al., 2021)), ensembling multiple modules (e.g., R2-D2 (Fajcik et al., 2021)), or creating multiple training loops to update the indexed documents multiple times (e.g., Hindsight (Paranjape et al., 2021)). In contrast, we focus on architectural efficiency improvements with a simple training paradigm. Recently, more task-independent retrieval-enhanced language models emerged, such as retrieving from a text-snippet database (Borgeaud et al., 2021) or learning to retrieve from the web with reinforcement learning (Nakano et al., 2021). For more information on retrieval-enhanced machine learning models, we refer the reader to Zamani et al. (2022).

**Improving and Adapting the FiD Model.** To integrate passage relevance prediction into FiD, Asai et al. (2021) add a second decoding module, which is called for every query-passage sequence to indicate its relevance. They also use this setup to generate silver-relevance scores for unjudged passages. Yu et al. (2022) replace the retrieval module with a large language model to generate supporting documents, which are then fused to generate the answer by a default FiD implementation. The current top-systems on the KILT leaderboard (Hofstätter et al., 2022; Izacard et al., 2022) use strong retrievers in combination with large T5-backbones for FiD. They also improve the supervised training by using better data sampling or pre-training procedures for more data efficient fine-tuning. We continue in the spirit of these related works with additional efficiency and capability improvements of FiD.

---

[2]Note we adapted the formulation of Lakhotia et al. (2021) from *sentence markers* to passage indices, to make the formulation more general.

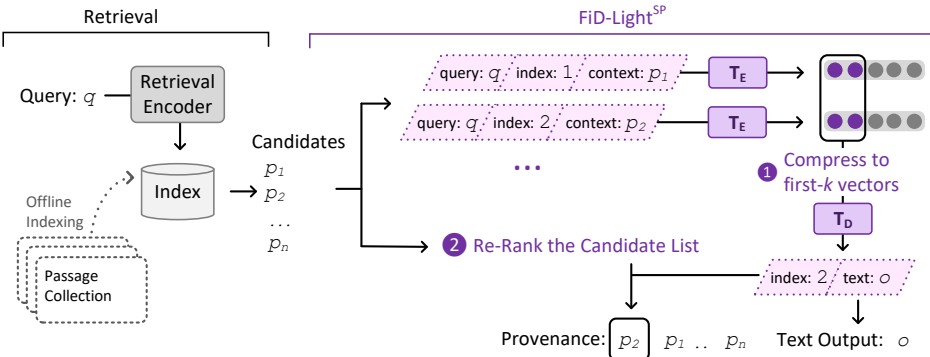

Figure 2: Overview of the FiD-Light architecture and workflow with source pointers. We highlight our two main contributions: ❶ Compressing the encoded vectors per passage, before concatenating and feeding them through the decoder; ❷ Increasing the robustness of source pointers, by using the model as re-ranker.

## 3  FID-LIGHT WITH SOURCE POINTERS

With FiD-Light[SP] we overcome the two main limitations of the FiD-Ex model and workflow: We drastically increase the efficiency of the decoder, by reducing its computational requirement, and we improve the robustness of the passage selection with a source pointing workflow, by shifting our view from an explanation to a second, parallel-solved task: re-ranking passages. We provide an overview of our FiD-Light[SP] model and source pointer workflow in Figure 2.

**Decoder Efficiency.**    Following our initial observation, that FiD spends most time in the decoding phase (Figure 1), we adapt the original FiD decoding step (Eq. 2) to reduce the length of each encoded query-passage pair to $k$ vectors via a function $f$:

$$\hat{o} = T_D([f_k(e_1); f_k(e_2); ...; f_k(e_n)]) \qquad (5)$$

This reduces the input length from the previous $O(n*(|q| + |p|))$ to $O(n*k)$, where $k \ll |q| + |p|$. The exact compression ratio depends on the required tokens for the used tasks; we experiment with configurations from a 6x to 384x fold reduction. In our experiments, for simplicity, we instantiate $f_k$ as the first $k$ vectors of each sequence. While this architecture change is simple, it strongly disrupts previous assumptions that every encoded token is accessible for decoding in the T5 architecture. Its simplicity also means that the community can easily adapt existing codebases with this change to benefit from the efficiency improvements.

**Source Pointing Robustness**    To enable source pointing in FiD-Light, we train the model with the source pointing concept proposed by Lakhotia et al. (2021) in FiD-Ex. Our novel contribution is how we handle the output of the source pointers at inference time. If we use them directly as result, as in FiD-Ex, we are prone to instability in the number of returned passages. The question of processing the output further almost becomes philosophical: if we treat the source pointers as explanations we can not process them any further without corrupting the explanation. While, there might be a correlation between the textual output and the source pointed passages, we are treating *finding the source passages* as a concurrent task to generating the text output. Because we are not claiming them to be explanations we can now process them further.

We propose to merge the initial ranked candidate list of passages $C$ with the source pointing selected passage by re-ranking the selected passages (found in the decoded output $\hat{o}$) to the top of the list:

$$\hat{C}_{1:r} = \left[ [r|r \in \hat{o}]; [r|r \in C, r \notin \hat{o}] \right] \qquad (6)$$

To compute all selected passages $r \in \hat{o}$ we first parse the output $\hat{o}$ with a simple parser for the trained format given in Eq. 4, including a conversion from the text-tokens representing the indices to integers. In case the model selects multiple passages we keep the selection order of the model output. If a task contains graded relevance annotations for training passages, we can train the model to follow the grades, if only binary relevance is available (as in the case with KILT), we keep the rank-ordering of the multiple selected passages from the initial candidate list. This change leads to higher robustness in our provenance results, as distribution differences between training and evaluation otherwise lead to a disadvantaged position, as we demonstrate in Section 4.2.

Table 1: FiD-Light downstream KILT-scores for different retrievers (realistic & oracle evaluation).

| | Open Domain QA | | | Fact | Slot Filling | | Dialog |
|---|---|---|---|---|---|---|---|
| **GTR Retriever** | NQ | HotpotQA | TriviaQA | FEVER | T-REx | zsRE | WOW |
| | *KILT-EM* | *KILT-EM* | *KILT-EM* | *KILT-AC* | *KILT-AC* | *KILT-AC* | *KILT-F1* |
| **Real-World Evaluation** | | | | | | | |
| 1   Zero-Shot | 38.0 ±.3 | 11.3 ±.2 | 30.8 ±.3 | 71.6 ±.4 | 64.9 ±.2 | 67.0 ±.6 | 9.5 ±.2 |
| 2   KILT Fine-Tuned | 41.4 ±.4 | 24.1 ±.2 | 37.6 ±.2 | 78.1 ±.2 | 71.5 ±.1 | 71.4 ±.4 | 10.9 ±.2 |
| 3   FT + Relevant (Train) | 40.3 ±.3 | 25.7 ±.1 | 37.5 ±.3 | 78.1 ±.3 | 71.7 ±.2 | 71.5 ±.6 | 11.4 ±.1 |
| **Oracle Evaluation** | | | | | | | |
| 4   FT + Rel. (Train&Eval) | 44.9 ±.6 | 45.9 ±.3 | 54.3 ±.4 | 83.0 ±.2 | 77.4 ±.2 | 75.2 ±.4 | 14.8 ±.2 |
| 5   Only Relevant | 63.9 ±.4 | 58.7 ±.3 | 78.6 ±.2 | 90.9 ±.1 | 90.6 ±.1 | 79.9 ±.3 | 21.8 ±.2 |

## 4   RESULTS

We empirically address the research questions laid out in the introduction. We study the importance of the retriever module, the efficacy of the source pointer workflow, the tradeoff between efficiency and effectiveness using a controlled baseline, and finally we compare our FiD-Light[SP] to related methods on the blind-evaluated KILT leaderboard. We detail our experiment design in Appendix A.

### 4.1   INFLUENCE OF THE RETRIEVER

The retrieval module is the backbone for all retrieval-augmented generation. The generation quality is to a large extent bound by the retrieval quality, especially if the retrieved information is not memorized by the generator. To answer **RQ1** *What impact does training the retrieval module have on FiD-Light[SP] downstream results?* we have to be careful to acknowledge the uncertainty of sparse ranking annotations (Hofstätter et al., 2022).

To accurately quantify the retriever's contribution, we compare the downstream effect of a zero-shot, a fine-tuned (methodology described in detail in Appendix B), and two oracle retrievers in Table 1. In the first section (rows 1-3) retrievers are evaluated without access to relevance judgements (a real-world environment), whereas in the second section (rows 4 & 5) we infuse relevance information during the evaluation (oracle environment). We find that training the retriever with in-domain training data (row 2) consistently improves results over a zero-shot retriever (row 1) as used by (Hofstätter et al., 2022). While always ingesting all known relevant passages during training (row 3) does not significantly change the downstream performance.

To account for annotation uncertainty in our *retriever as oracle* experiments, we study two scenarios: 1) infusing all known relevant passages into the retrieved candidate list (row 4) and 2) setting the candidates to be only the known relevant passages (row 5). Commonly, the community compares their results only against the second oracle scenario, showing a large headroom for future improvements for the retriever (Glass et al., 2021; Shuster et al., 2021). However, we argue, due to the sparsity of the annotations, we should compare the results to our more realistic first oracle scenario (row 4). It still shows a significant opportunity for improvement, albeit the total headroom is roughly halfed across the board. Future work may explore more fine-tuning aspects, but we decide to select the simple fine-tuned retriever (row 2).

### 4.2   SOURCE POINTER ROBUSTNESS

While the initial source pointer concept has been proposed by FiD-Ex as sentence markers for explainability, we are the first to study their application in the more complex passage ranking context combined with our compressed FiD-Light architecture. Therefore, we study **RQ2** *How robust is our source pointing and re-ranking workflow applied to FiD and FiD-Light?*

As introduced earlier, we train the source pointing capabilities into FiD(-Light) by flagging all known relevant passages retrieved in the candidate passage set. By directly using the size of the known relevant item set during training we instill a strong expectation prior into the model of how many passages ought to be relevant for a given task. Note, if a known relevant passage is not retrieved we cannot use it for training the generator. In Figure 3, we observe these effects for four representative tasks of the KILT benchmark. Each of these tasks shows a different expected distribution target. We note that the training distribution differs from the target, as it skips non-recalled relevant items. We find the model output distribution on the validation set to closely match the training distribution (albeit here we make no claims about the correctness of the selected passages).

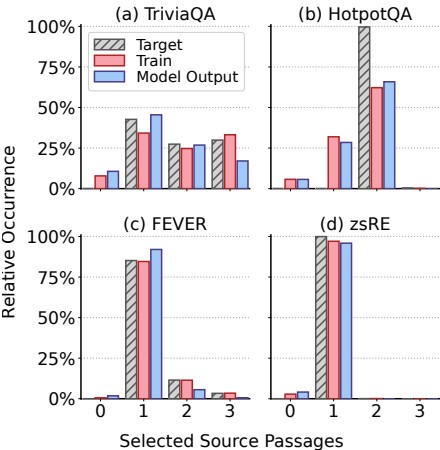

Figure 3: Distributions of source pointer passages for FiD-Light$^{SP}$ (T5-Base).

Table 2: Comparing our source pointer ($^{SP}$) re-ranking with the direct model output (Ex) using KILT scores for passages and documents. *Bold indicates improvement of SP over Ex larger than the 95% CI.*

| Model | Open Domain QA | | | | Fact | | Slot Fill. | |
| | HotpotQA | | TriviaQA | | FEVER | | zsRE | |
| | *Pas.* | *Doc.* | *Pas.* | *Doc.* | *Pas.* | *Doc.* | *Pas.* | *Doc.* |
|---|---|---|---|---|---|---|---|---|
| **T5-Base** | | | | | | | | |
| 1 FiD-Ex | 25.4 | 25.6 | 22.0 | 34.1 | 70.1 | 77.2 | 70.1 | 71.6 |
| 2 FiD$^{SP}$ | 25.8 | **26.1** | **23.1** | **39.5** | **71.1** | **78.3** | 70.1 | 71.7 |
| 3 FiD-Light-Ex | 23.5 | 23.7 | 18.8 | 32.1 | 70.0 | 77.1 | 69.3 | 71.2 |
| 4 FiD-Light$^{SP}$ | 23.8 | 24.1 | **19.8** | **37.6** | **71.6** | **78.1** | 69.3 | 71.4 |
| **T5-Large** | | | | | | | | |
| 5 FiD-Light-Ex | 26.6 | 26.9 | 22.6 | 36.3 | 72.6 | 79.2 | 70.9 | 72.7 |
| 6 FiD-Light$^{SP}$ | 26.9 | 27.3 | **23.5** | **41.4** | **74.2** | **80.4** | 70.9 | 72.8 |
| **T5-XL** | | | | | | | | |
| 7 FiD-Light-Ex | 28.2 | 28.4 | 24.8 | 38.7 | 73.9 | 80.5 | 73.1 | 75.9 |
| 8 FiD-Light$^{SP}$ | 28.4 | 28.7 | **25.7** | **43.8** | **75.5** | **81.7** | 73.2 | 76.1 |

However, focusing on higher passage counts in Figure 3 (a) TriviaQA and (c) FEVER shows that the model struggles to output 3 passages as often as it is expected to do. This weakness becomes visible, when we evaluate the standard R-Precision of the selection, which needs at least R returned items to reach the full score, given R known relevant items.

To overcome this limitation, we propose instead of directly outputting the selection (FiD-Ex) to move the selected passages to the top of the ranked list. This essentially transforms FiD(-Light) into a re-ranking model. In Table 2, we show the ablation study to confirm the usefulness of the proposed re-ranking on final downstream results. Our approach is strictly positive or neutral for the results, as we are filling up holes, that would result in penalties. Confirming our hypothesis originating in Figure 3, we see stat. significant improvements across all configurations on the two task, where the model struggled to fill up the full distribution: TriviaQA and FEVER.

While in this work we do not change the KILT evaluation methodology and optimize our models towards the current standard evaluation, we note that these findings represent interesting avenues for future work requiring evaluation setup changes: We may choose to train the model to only select a single passage or even re-rank the whole list with our textual source pointers as re-rankers.

We might be tempted to directly compare the inter-setting results in Table 2, for example FiD$^{SP}$ in row 2 with FiD-Light$^{SP}$ in row 4 (T5-Base). Here we observe, especially on HotpotQA and TriviaQA, a quality reduction, which would lead us to the conclusion that source pointing in FiD-Light is less robust than FiD. To put these results into perspective, we exemplary selected HotpotQA and plot the query latency as well as the R-Precision of the models in Figure 4. For FiD$^{SP}$, we modulate the number of input passages; for FiD-Light we modulate the number of vectors $k$ fed to the decoder and the backbone size. We clearly observe a stark reduction in quality for the FiD$^{SP}$ model, when the number of input passages is reduced. On the other hand our FiD-Light$^{SP}$ variants are able to almost keep the same level of effectivness, and larger backbones, while still faster than the FiD$^{SP}$ baseline also produce a higher quality. Therefore, an equal-efficiency comparison in Table 2 involves row 2 and row 8 (using T5-XL). We are diving deeper in these tradeoffs in the next section.

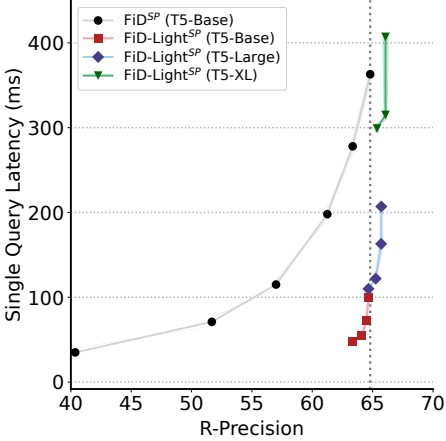

Figure 4: Comparing the capability to select the two relevant passages in HotpotQA for FiD-Light$^{SP}$ and FiD$^{SP}$.

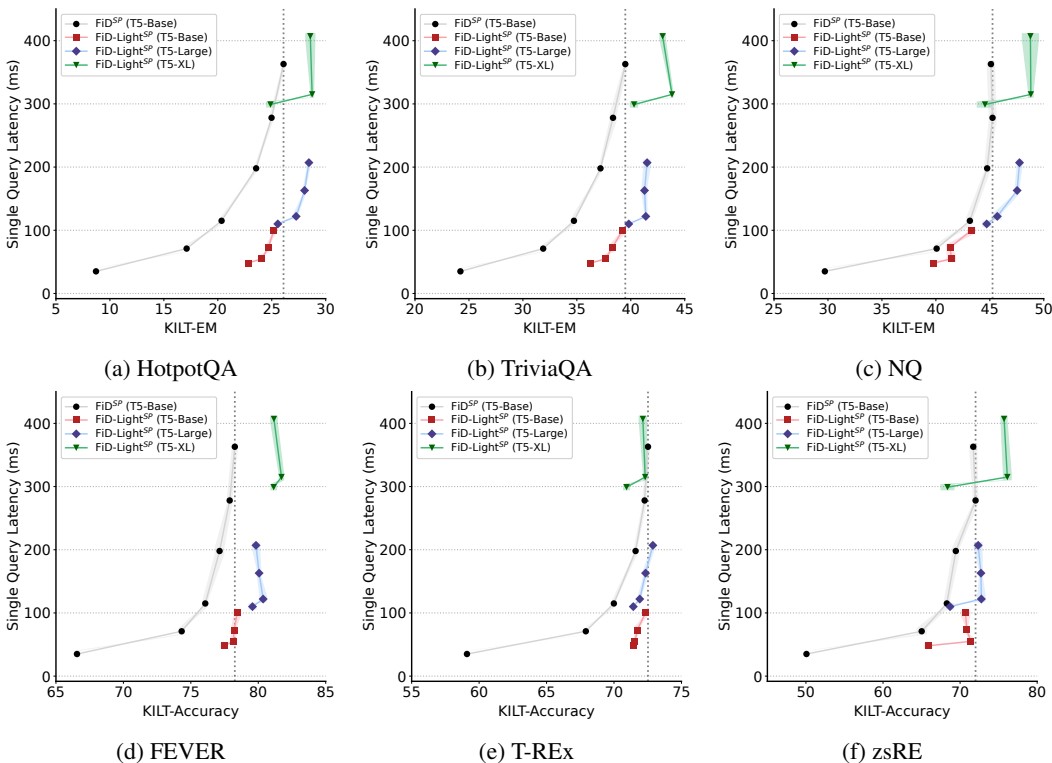

Figure 5: Comparing FiD-Light[SP] with FiD[SP] on KILT-scores modulating the number of input passages on FiD and the number of decoder-input vectors on FiD-Light.

### 4.3 EFFICIENCY - EFFECTIVENESS TRADEOFF

Ultimately, we as a community want our research be applied to real world use, to benefit society. A major component, besides concerns about safety and social biases as summarized by Bender et al. (2021), is the efficiency of the deployed system. To understand the impact of our proposed FiD-Light architecture we study **RQ3** *How does FiD-Light[SP] compare to the FiD[SP] baseline in efficiency-effectiveness tradeoffs?*

The KILT benchmark gives us the opportunity to study our changes in a large variety of tasks, with different properties, so that we can make confident claims about the efficacy of our changes. In Figure 5 we show our ablation results per task. For each task we report the average query latency (y-axes) and the main KILT-score effectiveness metric (x-axes). The gray line indicates our FiD baseline by modulating input passage counts – from 40 down to 1. Our FiD-Light models all have access to the full 40 passages, and here we are modulating T5 sizes as well as the number of vectors (1, 8, 32, 64) fed into the decoder.

We start our discussion with the open domain QA tasks in Figure 5 (a, b, & c) as they provide a similar picture: Comparing our FiD-Light[SP] model with the baseline we do observe a drop in effectiveness from the strongest baseline (gray dotted vertical line) when using the same T5-Base model. However, due to the more efficient architecture we are able to swap backbones and earn the benefits of those larger models in terms of effectiveness. At the same time we outperform the latency of the baseline as well, shifting the Pareto optimum. Interestingly, the FiD-Light[SP] model with T5-XL and only a single encoded vector per passage shows a larger drop in effectiveness than the counterparts for smaller T5's. The only 2-label classification task, FEVER, shown in Figure 5 (d), exhibits the lowest reduction in effectiveness, when constraining the number of encoded vectors in FiD-Light[SP]. This is likely due to the fact, that only little generation is necessary to solve the task. Therefore, our FiD-Light[SP] configurations improve the Pareto optimum again. The slot-filling tasks in Figure 5 (e & f) show less impact of the T5 size, with little improvement for Large and XL over the Base configurations. Fortunately, we also observe a similarly small reduction in effectiveness for reducing the number of encoded FiD-Light[SP] vectors, leading to our final Pareto gains.

In conclusion we observe clear and statistically significant improvements between FiD[SP] and FiD-Light[SP] – both in terms of effectiveness and efficiency – across a variety of KILT tasks. FiD-Light[SP]

Table 3: Comparing our models with related work on the KILT test set via the leaderboard (as of September 21, 2022). Highest result in bold; improvement over prior state-of-the-art underlined.

| | Model | Open Domain QA | | | Fact | Slot Filling | | Dialog |
|---|---|---|---|---|---|---|---|---|
| | | NQ | HotpotQA | TriviaQA | FEVER | T-REx | zsRE | WOW |
| | | *KILT-EM* | *KILT-EM* | *KILT-EM* | *KILT-AC* | *KILT-AC* | *KILT-AC* | *KILT-F1* |
| **Top Leaderboard Entries** | | | | | | | | |
| 1 | RAG (Petroni et al., 2021) | 32.7 | 3.2 | 38.1 | 53.5 | 23.1 | 36.8 | 8.8 |
| 2 | DPR + FiD (Piktus et al., 2021) | 35.3 | 11.7 | 45.6 | 65.7 | 64.6 | 67.2 | 7.6 |
| 3 | KGI (Glass et al., 2021) | 36.4 | – | 42.9 | 64.4 | 69.1 | 72.3 | 11.8 |
| 4 | Re2G (Glass et al., 2022) | 43.6 | – | 57.9 | 78.5 | 75.8 | – | 12.9 |
| 5 | Hindsight (Paranjape et al., 2021) | – | – | – | – | – | – | **13.4** |
| 7 | SEAL + FiD (Bevilacqua et al., 2022) | 38.8 | 18.1 | 50.6 | 71.3 | 60.1 | 73.2 | 11.6 |
| **Ours** | | | | | | | | |
| 8 | FiD-Light$^{\text{SP}}$ (T5-Base, $k = 64$) | 45.6 | 25.6 | 57.6 | 80.6 | 76.0 | 81.1 | 11.9 |
| 9 | FiD-Light$^{\text{SP}}$ (T5-Large, $k = 32$) | 49.9 | 28.2 | 61.4 | 82.1 | **76.7** | **84.1** | 12.2 |
| 10 | FiD-Light$^{\text{SP}}$ (T5-XL, $k = 8$) | **51.1** | **29.2** | **63.7** | **84.5** | 76.3 | 84.0 | 13.1 |

can lower the query latency by more than 2x and still deliver higher effectiveness by upgrading the language model backbone size.

## 4.4 COMPARISON TO RELATED WORK

In addition to showing improvements over our own baselines, we now demonstrate the effectiveness of FiD-Light$^{\text{SP}}$ in a broader context and answer **RQ4** *How does FiD-Light$^{\text{SP}}$ compare to related methods on the KILT benchmark?* The community is fortunate to have a blind-evaluation leaderboard for all KILT tasks[3] at our disposal to compare our approaches on a level playing field, where everyone may submit their highly-tuned systems. While the top spots of a leaderboard are typically not populated by efficient methods, we nevertheless submitted three different configurations of FiD-Light$^{\text{SP}}$ – all more efficient than our FiD baseline with 40 input passages. We selected a single checkpoint to submit for all tasks, so as to demonstrate our multi-task capabilities and not overfit a single submission to a single task.

We show the leaderboard results for the main KILT-score metrics in Table 3. For the independent breakdown of text generation and retrieval leaderboard scores we direct the reader to Appendix C. Even our T5-Base configuration in row 8 already outperforms previous SOTA results on five out of the seven tasks. With T5-Large and T5-XL (both continuously reducing the number of encoded vectors, to increase efficiency) set new SOTA results on six out of the seven tasks. Only WoW remains a weak spot, albeit not dramatically different to previous results. The fusion capabilities of FiD paired with our robust source pointing set especially impressive results on the challenging HotpotQA task, where exactly two distinct passages containing parts of the answer have to be placed on top of the ranked list. Here, we outperform previous methods by 61% or 11.1 KILT-EM points. On the other two QA task we reach +7.5 K-EM (+17.2%) for NQ and +5.8 K-EM (+10.0%) for TriviaQA. The zsRE task with +10.8 K-AC (+14.8%) and FEVER with +6.0 K-AC (+7.6%) round off our strong new SOTA results across a variety of tasks.

## 5 CONCLUSION

We proposed the FiD-Light model with a robust source pointing workflow to overcome efficiency and versatility limitations in the previous state-of-the-art retrieval-augmented generation model FiD. We adapted the FiD model architecture to compress the amount of information fed to the decoder, for drastically reduced inference latency. We demonstrated at the same time only a modest reduction in effectiveness, which can be alleviated with larger T5-backbones leading to Pareto optimal results on six KILT tasks. Our multi-task system achieved substantial new state-of-the-art results for combined retrieval and generation metrics on six KILT tasks compared to previous methods on the public leaderboard. These results demonstrate that we do not need to always scale up to achieve the highest effectiveness, enabling more researchers to work on this problem in the future.

---

[3]The leaderboard is available at: https://eval.ai/web/challenges/challenge-page/689

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

Table 4: KILT tasks grouped by category with training example and query set size statistics.

| Category | Dataset Name | Reference | Training # | Dev # | Leaderb. # |
|---|---|---|---|---|---|
| **Open Domain QA** | HotpotQA | (Yang et al., 2018) | 68,659 | 5,600 | 5,569 |
| | TriviaQA | (Joshi et al., 2017) | 177,238 | 5,359 | 6,586 |
| | Natural Questions (NQ) | (Kwiatkowski et al., 2019) | 89,372 | 2,837 | 1,444 |
| **Slot Filling** | T-REx | (Elsahar et al., 2018) | 197,439 | 5,000 | 5,000 |
| | Zero Shot RE (zsRE) | (Levy et al., 2017) | 137,945 | 3,724 | 4,966 |
| **Fact Verification** | FEVER | (Thorne et al., 2018) | 83,141 | 10,444 | 10,100 |
| **Dialog** | Wizard of Wikipedia (WoW) | (Dinan et al., 2018) | 54,330 | 3,054 | 2,944 |

## A EXPERIMENT DESIGN

**Implementation.** Our experiment setup follows the state-of-the-art multi-task relevance sampled training sets of Hofstätter et al. (2022). All our experiments are based on the T5X framework (Roberts et al., 2022). We start with a GTR-Base dense retrieval model (Ni et al., 2021), which is pre-trained on the MSMARCO passage retrieval task (Bajaj et al., 2016) and has been shown to generalize well on the BEIR benchmark (Thakur et al., 2021). We train our FiD(-Light) models using T5 v1.1 as language model backbone (Raffel et al., 2020) on TPUs. We attach task-specific markers to the queries for the multi-task training. We cap the input at 384 tokens (combined query and passage) and a maximum of 64 output tokens. For training, we use a batch size of 128 with up to 40 retrieved passages, and a learning rate of $10^{-3}$ with the Adafactor optimizer (Shazeer & Stern, 2018). We do not tune our models to a specific checkpoint, rather train them all for 50K steps. The only special case is T5-XL, which uses a learning rate of $5 * 10^{-4}$ and is trained for 30K steps. During decoding we use beam search with a beam size of 4.

**Datasets.** We conduct experiments on 7 KILT tasks: HotpotQA (Yang et al., 2018), TriviaQA (Joshi et al., 2017), Natural Questions (NQ) (Kwiatkowski et al., 2019), T-REx (Elsahar et al., 2018), Zero Shot RE (zsRE) (Levy et al., 2017), FEVER (Thorne et al., 2018), and Wizard of Wikipedia (WoW) (Dinan et al., 2018). We give an overview over the dataset in Table 4. We used the filtered training & passage sets from Hofstätter et al. (2022) and the original evaluation sets from Petroni et al. (2021).

**Evaluation.** We follow the KILT evaluation setup proposed by Petroni et al. (2021), in particular we focus on the main KILT-score metrics, which combines both a text output metric $M$ (such as EM, Accuracy, or F1) with R-Precision ($RP$) per query, before aggregating the individual query results over the query result set $Q$:

$$K_M = \frac{1}{|Q|} \sum_{q \in Q} M(q_{\text{text}}) * (RP(q_{\text{provenance}}) == 1) \qquad (7)$$

In essence, KILT-scores only count the text score $M$ if the R-Precision of the query is 1, meaning all $R$ relevant passages or documents are returned on the top-$R$ positions of the ranked list. This metric makes the assumption that only a few (1 to 2) items are marked as relevant, as is the case in the KILT dataset. To reduce the noise in our dev results, we present the mean and a 95% confidence interval measured with a t-statistic of the last 10 checkpoints (every thousand steps from 40K to 50K training steps). For our leaderboard submission, we selected a single checkpoint for all tasks. Unfortunately, we can not compute statistical significance tests compared to other methods, as the submission files and gold-labels are not publicly available.

Table 5: Passage-level Recall@40 KILT dev results of the GTR dense retrieval model with an additional relative difference to the respective Recall@40 on the training set ($\Delta$ T) – a lower difference is better.

| | Open Domain QA | | | | | | Fact | | Slot Filling | | | | Dialog | |
| Retrieval Model | NQ | | Hotpot | | TriviaQA | | FEVER | | T-REx | | ZS-RE | | WOW | |
| | R@40 | $\Delta$ T | R@40 | $\Delta$ T | R@40 | $\Delta$ T | R@40 | $\Delta$ T | R@40 | $\Delta$ T | R@40 | $\Delta$ T | R@40 | $\Delta$ T |
|---|---|---|---|---|---|---|---|---|---|---|---|---|---|---|
| Zero-Shot | 0.73 | 3% | 0.52 | -1% | 0.48 | 1% | 0.84 | 4% | 0.75 | -1% | 0.89 | 0% | 0.58 | -3% |
| Trained LR: 0.1 | 0.83 | -17% | 0.70 | -33% | 0.64 | -39% | 0.80 | -16% | 0.84 | -7% | 0.86 | -9% | 0.73 | -34% |
| Trained LR: 0.05 | 0.85 | -13% | 0.73 | -28% | 0.66 | -33% | 0.88 | -11% | 0.86 | -5% | 0.89 | -7% | 0.73 | -33% |
| Trained LR: 0.01 | 0.86 | -6% | 0.72 | -12% | 0.66 | -15% | 0.93 | -5% | 0.88 | -2% | 0.92 | -5% | 0.75 | -13% |
| Trained LR: 0.005 | 0.85 | -3% | 0.72 | -7% | 0.66 | -9% | 0.94 | -4% | 0.88 | -1% | 0.93 | -4% | 0.75 | -8% |
| Trained LR: 0.001 | 0.83 | -1% | 0.69 | -2% | 0.64 | -2% | 0.94 | -2% | 0.88 | -1% | 0.94 | -4% | 0.75 | -4% |

# B    DENSE RETRIEVAL TUNING RESULTS

In our experiments we use a "double-finetuned" GTR dense retriever retriever: First it was trained on the MSMARCO retrieval task (Bajaj et al., 2016) by Ni et al. (2021) and then we fine-tuned their checkpoint further on our combined KILT training set to create a single generalized KILT retrieval module, akin to Maillard et al. (2021). We created passage retrieval training triples containing a query, a known relevant passage, and a sampled negative passage (randomly sampled from the top-100 GTR zero-shot rankings for the query). We then fine-tuned the retriever for 100K steps using the GTR default parameters in the t5x_Retrieval framework. We did not employ knowledge distillation (Hofstätter et al., 2020) or complex end-to-end losses (Izacard et al., 2022), to demonstrate the effectiveness of our approach in a simple setting which likely is orthogonal to more complex training setups.

This approach means, that while we expect to learn retrieve better results, we may overshoot our target and overfit on the training data, leading to a growing divide in the the train vs. test performance. This matters strongly in our retrieval-augmented generation setup, because we use the fully trained retrieval model as the source for our generation training data. We aim to detect and avoid unnecessary distribution shifts to actually train the generator on the expected retrieval performance and not an overfitted training set.

We choose to modulate the learning rate to control for and study the train vs. test distribution shift. We focus on the recall at the highest cutoff we use in our experiments (the top-40) and provide our results in Table 5. First, we show the zero-shot results, as used by the previous state-of-the-art FiD models from Hofstätter et al. (2022), followed by our novel fine-tuned GTR models. Our first observation is that in all tasks we are able to achieve significant R@40 gains on the dev set compared to the zero-shot baseline – ranging from 0.13 to 0.20 absolute changes. Concerning our learning rate study, we find too high learning rates (especially 0.1 and 0.05) show a high $\Delta$T, which indicates a strong distribution shift between train and test. If we were to only train one of the high learning rate checkpoints and compare the dev results to the zero-shot baseline we could be tempted to use them, as their dev results look strong. However, due to our fine-grained analysis we see that it would introduce a strong distribution shift.

Another interesting observation we make is how different task categories seem to converge at different velocities – the open domain QA tasks reach their optimal dev results with higher learning rates, while the other tasks fare better with lower rates. Curiously, we would have guessed a reverse trend, as the initial MSMARCO retrieval task is more closely aligned to QA, suggesting less needed movement. We did not continue to tune the composition of our retrieval training as it is only a secondary contribution to this work and the differences are quite small compared to the margin we achieve to the zero shot baseline. Therefore, we decided to go forward with the 0.005 learning rate, as it overall gives the best results with low distribution shifts.

Table 6: Comparing our models with related work on the KILT test set via the leaderboard (as of September 21, 2022). Highest result in bold; improvement over prior state-of-the-art underlined.

| Model | Open Domain QA | | | | | | Fact | | Slot Filling | | | | Dialog | |
| | NQ | | HotpotQA | | TriviaQA | | FEVER | | T-REx | | zsRE | | WOW | |
| | EM | R-P | EM | R-P | EM | R-P | AC | R-P | AC | R-P | AC | R-P | F1 | R-P |
| **Top Leaderboard Entries** | | | | | | | | | | | | | | |
| 1  RAG (Petroni et al., 2021) | 44.4 | 59.5 | 27.0 | 30.6 | 71.3 | 48.7 | 86.3 | 61.9 | 59.2 | 28.7 | 44.7 | 53.7 | 13.1 | 57.8 |
| 2  DPR + FiD (Piktus et al., 2021) | 51.6 | 59.8 | 36.9 | 47.2 | 72.7 | 59.7 | 89.0 | 74.8 | 81.3 | 75.6 | 74.0 | 89.6 | 15.7 | 41.5 |
| 3  KGI (Glass et al., 2021) | 45.2 | 63.7 | – | – | 61.0 | 60.5 | 85.6 | 75.6 | 84.4 | 74.4 | 72.6 | **98.5** | 18.6 | 60.1 |
| 4  Re2G (Glass et al., 2022) | 51.7 | 70.8 | – | – | 76.3 | 72.7 | 89.6 | 88.9 | **87.7** | 80.7 | – | – | 18.9 | 60.1 |
| 5  Hindsight (Paranjape et al., 2021) | – | – | – | – | – | – | – | – | – | – | – | – | 19.2 | 56.1 |
| 7  SEAL+FiD (Bevilacqua et al., 2022) | 53.7 | 63.2 | 40.5 | 58.8 | 70.9 | 68.4 | 89.5 | 81.5 | 83.7 | 67.8 | 74.7 | 98.0 | 18.3 | 57.6 |
| 8  FiD with RS (Hofstätter et al., 2022) | 61.2 | – | 39.1 | – | **84.6** | – | 92.3 | – | 85.2 | – | 83.7 | – | 20.6 | – |
| 9  FiD with Atlas (Izacard et al., 2022) | **61.3** | – | **50.6** | – | 84.0 | – | **93.5** | – | 85.1 | – | 80.8 | – | **21.6** | – |
| **Ours** | | | | | | | | | | | | | | |
| 10  FiD-Light^SP (T5-Base, $k=64$) | 52.6 | 71.5 | 37.1 | 65.6 | 73.3 | 72.8 | 89.3 | 91.4 | 85.5 | 81.5 | 81.9 | 96.1 | 16.5 | 62.8 |
| 11  FiD-Light^SP (T5-Large, $k=32$) | 57.3 | 74.2 | 40.2 | 66.6 | 77.1 | 74.4 | 90.9 | 91.4 | 86.4 | **81.7** | 84.9 | 96.2 | 16.9 | 65.5 |
| 12  FiD-Light^SP (T5-XL, $k=8$) | 58.4 | **75.5** | 42.5 | **67.2** | 80.1 | **74.7** | 92.9 | **92.1** | 86.2 | 81.4 | **85.3** | 96.1 | 17.8 | **66.1** |

## C  DETAILED RELATED WORK COMPARISONS

In Table 3 we focused on the combined retrieval and text generation KILT scores. Now, we investigate our results further, by analyzing the two components independently in Table 6. For each task we report the leaderboard text generation test score (EM, AC, or F1) and the retrieval quality via R-Precision. As previously noted, (Izacard & Grave, 2020; Hofstätter et al., 2022), there is a strong correlation between model size and text generation quality on KILT. For better comparability, and to not "poison" the task with only very large models, that are not trainable for many of our fellow researchers, we report small and large model numbers for FiD-Light.

Looking at the existing leaderboard entries we observe the top systems mostly rely on the FiD architecture. The most recent and highest performing approaches are FiD generators with relevance sampling and Atlas training regimes (row 8,9). It is important to note, that these two systems are very inefficient: They run 50 and 100 passages through FiD per query and use T5-XL and T5-XXL respectively. They also only focus on the text generation part of the KILT challenge, and chose not to submit any supporting passages for the generation. This is in large part due to the fact, that FiD on its own does not provided a ranking component to the passages, which leads to under-performing results.

Our FiD-Light^SP entries cover multiple T5 and $k$ encoded vector sizes. While there is our expected spread of the text generation quality based on the the T5 size, we observe that this spread is substantially smaller for the R-Precision metric. To be able to compare methods, the KILT leaderboard computes the R-Precision on a document level. We transformed our passage ranking to document ranks, by taking the highest ranked passage per document as the document rank, and removing subsequent passages from that document from the ranked list. Overall all our models beginning with T5-Base set new SOTA results across the board for the ranking sub-task, even considering we only re-rank 40 passages. Analysing the text generation quality, we see no new SOTA results for FiD-Light^SP, but we remain competitive with the largest and slowest entries in the leaderboard.

To conclude, we showed the reason for our overall strong SOTA results on the KILT scores in Table 3 is the combination of competitive text generation quality with strong SOTA ranking results shown in Table 6.

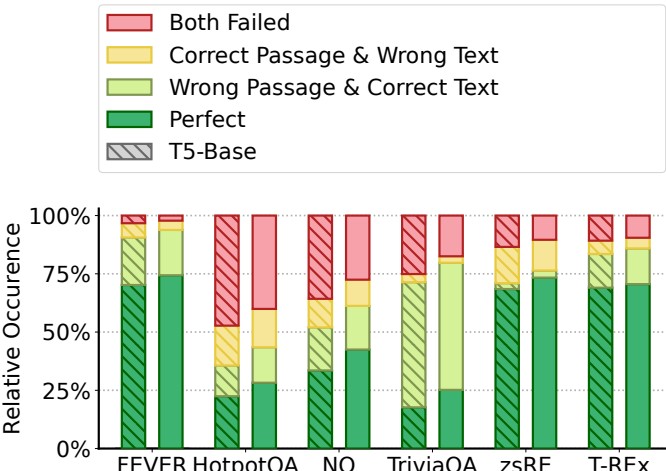

Figure 6: Failure type analysis on KILT dev sets per task for FiD-Light[SP] with T5-Base (hatched) and T5-XL (plain) backbones.

## D  FAILURE ANALYSIS

The setup of the knowledge intensive text generation with supporting passages, not only enables positive evaluation via the KILT scores, but also a rich quantitative failure analysis. As Boyd-Graber & Börschinger (2019); Hofstätter et al. (2022) argued, we should spend more time and energy looking beyond our aggregated metrics. Therefore, in Figure 6 we look at the composition of the raw output results of FiD-Light[SP] (without re-ranking) in 4 potential outcomes: 1) both passage and text results are wrong; 2) correct passage, but wrong text; 3) correct text, but wrong passages; and 4) both result parts are correct. We analyze the results of two T5-backbones across our KILT tasks.

Interestingly, we do not observe converging trends in their failures between the Base and XL backbones across tasks. But we do see strong differences in the distribution of failure types between tasks. The open domain QA tasks are more likely to fail, especially both parts. For the FEVER fact verification, if we scored the relevant passage on top we are very likely to also get the right boolean answer. The large part of wrong passage selection, but right answer in TriviaQA is likely attributed to its high degree of noise as observed by Hofstätter et al. (2022). HotpotQA remains the most challenging task with the highest double failure rate.

We note that the KILT tasks are highly noisy: we only have 1-2 relevant marked passages in most cases and few if any textual variations of the text answers. This is also the reason we did not run this analysis on WoW, which has no exact text matches. We hypothesize, that if both result parts fail, we are more likely to have a true failure of the model compared to only failing one aspect, which could indicate a noise issue in the datasets. However, to confidently claim this we would need to conduct a thorough annotation campaign of the existing results.

We created an interactive website for inspecting all model outputs of FiD-Light split by our failure analysis modes from Figure 6. The website displays the user 10 random results, per category and task, so as not to enable cherry picking by us. Every refresh of the website creates a new random sample allowing the users to playfully, yet targeted explore the datasets and results in a qualitative way. The website is available at: anonymized

