# OpenReview forum: "FiD-Light: Efficient and Effective Retrieval-Augmented Text Generation"
_ICLR.cc/2023/Conference — Submitted to ICLR 2023_

### Official Review · Reviewer_erWw · 2022-10-24

**Confidence:** 4
**Correctness:** 3
**Technical Novelty And Significance:** 2
**Empirical Novelty And Significance:** 2
**Recommendation:** 6

**Clarity, Quality, Novelty And Reproducibility:**

The paper is often not easy to read, due to somehow dense manner. A substantial revision for readability may be required for better clarity. The proposed idea is an incremental work of FiD, not being very novel (not so innovative), although it is helpful to the litererature. The experiments of the tradeoff on the efficiency and effectiveness are solidly done, demonstrating the interesting behaviors and results of FiD and FiD-Light.



**Strength And Weaknesses:**

Strengths
- This proposed idea is quite simple and well-motivated, making an interesting extension of FiD, which would be helpful to the FiD-based literature.
- The experiment results are solidly done, showing that the proposed FiD-Light is quite efficient, as well as keeping the effectiveness on various datasets.

Weaknesses:
- The proposed method mainly focuses on the efficiency. While single query latency is used for the efficiency metric in the paper, FiD could be designed efficiently under the batch-style GPU-based parallel processing. Thus, a stricter GPU-aware time complexity needs to be used for the efficiency measure of FiD. The concern is that the time complexity of FiD may be exaggerated in the experiments. Assuming the realistic situation that the input is processed at the maximum length limit of the encoder using the nicely designed GPU-based implementation, the authors need to compare the time complexities between the proposed method and FiD.
- In the proposed method, the chosen options need to be further explored or discussed. Why the first-k tokens in FiD-Light are selected? Why the source pointed passages are moved forward? (not moved backward)? How performances are changed when varying k?
- The naïve combination of reranking and FiD can reduce the time complexity, in the way that after reranking, the ranked top m passages can be selected as input for FiD while discarding other passages. But, it is not clear whether these reranking-and-selecting experiments are performed in the comparison of the paper.



**Summary Of The Paper:**

This paper proposes an extension of Fusion-in-decoder for improving both the effectiveness and efficiency. To improve the efficiency, noting that the decoding step occupies most of the time complexity, the method proposes a “selection-based compression” of the encoded representations, denoted as FiD-Light, by using only first-k vectors in the sequnce of the encoded token representations for each passage before directly fusing them in the decoder. To enhance the effectiveness, the reranking is performed based on the modified application of the “source pointing” method, denoted as FiD^{SP} by FiD-Ex by explicitly moving the source-pointed passages (i.e. highly relevant passages) forward in the list of the passages in the encoder, such that the distribution of the encoder’s input becomes more similar between training and test samples. Experiment results show that the modified “source pointing” method of Fid-Ex improves the performance both on the setting of FiD-Ex and FiD-Light. In addition, FiD-Light^{SP}, the method of applying both FiD-Light and FiD^{SP}, substantially reduces the time complexity, demonstrating that FiD-Light^{SP} achieves significantly improved performance at the fixed latency time budget.

Overall, the key contribution of the paper is the simple but novel extension of FiD, which has widely been used for retrieval-augmented LM and its empirical usefulness on several standard datasets.



**Summary Of The Review:**


Overall, the paper is well-motivated with an interesting extension of the widely-used FiD, drastically improving the efficiency. The detailed experiments comparing with other recent SOTA works are also helpful to the literature.
While the major parts are helpful and quite interesting, however, major concerns are as follows:
1. the efficiency measure may not be fairly designed. GPU-based parallel processing needs be well-considered for designing FiD.
2. the baseline method of reranking needs be evaluated, in the way that the reranked passages are only selected as input for the encoder.

Also, the paper is often not very readable for some parts, so requiring a substantial revision for bettter clarity.

---

> ### Author Response · Authors · 2022-11-17
> **Author Response**
>
> Thank you for taking the time to read our paper and your constructive feedback! We answer your questions as follows:
>
> **[choice of f_k]** We acknowledge that many possible techniques for vector reduction could be used for f_k (windowed pooling, extra attention layer, etc..), therefore we formulated FiD-Light in an abstract way. We used the very simple first-k selection as it can be easily implemented by everyone who already uses FiD with a specific setup in a few lines of code. Furthermore, we wanted to show that our intuition of reduced vectors already bears fruit in a simple setup, without overengineering the approach. Future work may now build on our results to explore more options.
>
> **[source pointer re-ranking]** We move the selections to the top of the ranked list, as we train the model to select the most relevant passages and the ranked retrieval list is evaluated from top to bottom = best to lowest relevance.
>
> **[ablation of k parameter]** We show the ablation of the k parameter in Figures 4 & 5: the different points per FiD-T5 instance are different settings of k.
>
> **[GPU parallel processing]** We already make use of parallel processing as much as possible: The passages are encoded in parallel (even with a single query), however, the autoregressive nature of the decoding requires this substantial time as we need to re-evaluate every layer per newly generated token. We used a query batch size of 1 for the **latency** measurements, as this is the fastest possible way to compute a single query. We may be able to increase the **throughput** with batched queries (as we do during training), but in this context, we focus on the latency as this is the main efficiency metric we are concerned with in our online compute setting (as we can not pre-compute answers in advance).
>
> **[re-ranking baseline]** We agree that not including these additional “re-ranking before FiD” baselines is the biggest weakness in our paper. Unfortunately, we ran out of time and space in the paper to include them as well. What we do have is an ablation on our initial retriever quality (zero-shot vs trained) where we observed that both FiD and FiD-Light moved in tandem to improve the downstream results with the better retriever. If we now add another re-ranker before FiD, we could lower the cost of FiD, but also of FiD-Light, therefore the efficiency benefit would persist in any case. Furthermore, if we look towards related methods, such as the recent Re2G model, which uses a telescoped retrieval->re-ranking->generation approach, we do not see an effectiveness benefit of adding a re-ranker to the system, even for retrieval alone, as we strongly outperform Re2G on R-Precision on 6 KILT tasks (see Table 6 - R-Precision columns).
>
> A final ask: could you point us to specific passages/sections you would like us to polish and improve the readability? That would help us a lot to further improve our paper in the future. Thank you!
>
> We hope we could answer your questions and look forward to your response. Thank you again for reviewing our paper!

---

### Official Review · Reviewer_aESm · 2022-10-25

**Confidence:** 4
**Correctness:** 3
**Technical Novelty And Significance:** 2
**Empirical Novelty And Significance:** 3
**Recommendation:** 5

**Clarity, Quality, Novelty And Reproducibility:**

The novelty relies mostly on the use of a new ranking loss, but unfortunately this is the less well explained part of the paper (and this hinders reproducibility a lot, as well as the interest of the paper):

- in section 3: Eq. 6 is not understandable: $\hat o$ is supposed to ber the representation of the encoded passages (Eq. 5) - what does $r \in \hat o$ mean? Also, how is used when training the model? In the experimental section,
- How figure 3 shows that the model struggles to output 3 passages as often as it should? What does "we are filling up holes" means? How can we "observe a stark reduction when the number of input passages is reduced" on Figure 4?

In section 4.4, table 3 should be replaced by the one in appendix (non aggregated results) since the picture is less clear.

Other parts that could be improved are the related works section (no positioning of the paper with respect to related works).


**Strength And Weaknesses:**

Strengths: simple modification of the FID model that improves its performance a lot.

Weaknesses: lack of clarity in the description of the model (still not quite sure what the source pointing method is, but the answers clarified y doubts)

The proposed modifications are well-motivated and sensible, showing an improvement for the KILT tasks. Both modifications are quite simple but do work in practice, the paper has some experimental value.



**Summary Of The Paper:**

This paper modifies two aspects of the FID model (retrieval-augmented text generation) in section 3: (1) the authors truncate the passages to speed up the model (2) they modify the explainability component by using a ranking task. Results on KILT show a substantial improvement over the FID model.


**Summary Of The Review:**

This paper proposed a modification of an existing model (FID) that improves its efficiency with a simple truncation of inputs -- and a modification of a loss (but which is not well explained in the paper). Overall, the novelty is quite limited nevertheless and the paper lacks of clarity in many parts.

---

> ### Author Response · Authors · 2022-11-17
> **Author Response**
>
> Thank you for taking the time to read our paper and your constructive feedback! We answer your questions as follows:
>
> **[novelty concerns]** We agree that our algorithmic contributions are quite simple. We would however argue that this simplicity should not be confused with lack of novelty or potential impact. The simplicity - and the fact that it works, as shown by our many empirical studies - is one of the main strengths of our work. Currently, the community working on retrieval augmented generation models is composed mostly of corporate researchers with access to large amounts of compute, simply because the SOTA model FiD is so costly to train and evaluate. This is in stark contrast to the information retrieval and larger NLP communities, where many more academic researchers are able to participate with modest resources in a variety of tasks. By introducing simple modifications that can be easily adapted into existing codebases, we aim to open up the field of retrieval augmented generation to a larger community.
>
> **[source pointing - overview]** We believe there might be a few misunderstandings. **We are not proposing a new ranking loss and we are not changing the training aspects of the source pointers proposed in FiD-Ex**. Our contribution is changing the way we handle the output during inference - to make it more robust to improve the results. As Eq. 5 states o^  is the decoded output and not the encoded passage representation. (Eq. 5 calls T_D (the T5-Decoder)). o^ is a list of tokens (equivalent to a plain string, by re-mapping the vocabulary). Eq. 6 is now only concerned with the inference output handling and not the training itself. The training is unchanged from FiD-Ex, where we optimize the cross-entropy of the literal text tokens corresponding to the selected passage indices with the rest of the output text. In Eq. 6 r∈o^ is a shorthand for: the literal token for r (the number 1-40) is part of the start of the output string o^. This is accomplished with a simple parser written in pure python operating on the plain output text, we parse the output by “index:” and “text:” markers, followed by the number parsing. The parsed numbers are corresponding to the passage indices which will be re-ranked from the candidate list. If the parse fails at any point, we fall back to not selecting any passages (although this never happens).
>
> **[source pointing - analysis]** In Figure 3 we plot the number of passages output by the model, the number of known relevant passages in the dataset, and the number of passages known during training (only those that have been recalled by the retriever). For FEVER and TriviaQA we can see that the blue bar (the model output) is lower than the hatched bar for the expected target for 2 and 3 passages respectively. These are the passages that are missing from the model output. Unfortunately, due to space constraints the differences look quite small, but there is actually an around 10% mismatch of the total distribution of passages. In those 10% of cases we can not reach 1 for the retrieval quality metric R-Precision - we are handicapped by the limits of FiD-Ex. Therefore we propose to “fill up” this hole by re-ranking the candidate list, to always have all candidate passages in the result set. These 10% are then the reason for the 1 to 5 point gap between the FiD-Ex baseline and our SP re-ranking method observed in Table 2. The analysis in Figure 4 is concerned with a follow up analysis on the number of passages to even re-rank, where the output handling is solely based on the SP method of re-ranking (as the labels in the Figure indicate).
>
> **[choice of main result metrics]** We follow the original KILT paper by Petroni et al (https://arxiv.org/pdf/2009.02252.pdf) and report their main KILT effectiveness metric, as these have been proposed specifically for the KILT benchmark. The default nature of the KILT scores ist further emphasized by the fact that the official leaderboard uses the KILT-EM/AC/F1 scores as the default ranking for leaderboard entries. To provide further insight into the models we also appended the individual metrics in the Appendix. Looking at the individual metrics, we observed that for the retrieval-quality R-Precision we strongly outperform previous methods, only on the text generation quality we show “mixed” results for the simple fact that the stronger related method Atlas uses far more compute (T5-XXL with 11B params). While direct comparisons are not possible, we likely outperform their efficiency by 1 order of magnitude and more.
>
> **[related work]** We mainly situated our work wrt. to related work in the introduction section as well as in the first of the three paragraphs of the related work section 2.2; Thank you for observing our missed entries. We also added a positioning of our paper in the remaining two paragraphs.
>
> We hope we could answer your questions and look forward to your response. Thank you again for reviewing our paper!

---

> > ### Comment · Reviewer_aESm · 2022-11-25
> > **Response**
> >
> > Thanks for your answers that clarified a lot of what you did - I still find the contribution interesting but not so novel but will increase my rating since some of my doubts are cleared.

---

### Official Review · Reviewer_aPe7 · 2022-10-25

**Confidence:** 4
**Correctness:** 3
**Technical Novelty And Significance:** 3
**Empirical Novelty And Significance:** 3
**Recommendation:** 6

**Clarity, Quality, Novelty And Reproducibility:**

The paper is clear and well-written. The appendix contains information for reproducibility.

**Strength And Weaknesses:**

Strength
+ simple and effective solution to improve efficiency
+ clear improvements across datasets and tasks

Weaknesses
- some parts of the paper should be better clarified (see my comments below)


**Summary Of The Paper:**

The paper introduces FiD-light, a more efficient variant of the fusion-in-decoder model that maintains/outperforms state-of-the-art performances on the KILT dataset, while drastically increasing the model's efficiency. To achieve this, FiD light compresses the length of input vectors and uses re-ranking to improve the top-ranked provenance precision.



**Summary Of The Review:**

Overall I found the paper clear. I do have some concerns regarding some choices made regarding the architecture and a couple of suggestions/questions.

1) Intro RQ2 - what's unexpected in distribution learned by FiD-light?
2) Sec 3 Decoder efficiency - what's the fk function used to compress vectors?
3) In some sections of the paper (e.g., sec 4.1) you are referring to "common practices" without citing papers that follow such approaches. For example, Sec 4.1 mentions that the community compares results to your second oracle scenario. I suggest adding citations to relevant works that do this.
4) Other works (e.g., KG-FiD) applies re-ranking between the encoder and the decoder of the T5 model. In this work, instead, you focus on the source pointing. I might have missed this part, but it is unclear why you are taking that direction. What's the reason for including source pointing?
5) Sec 4.3 mentions that the model lower the latency by 2x. It would be good to specify with respect to which of the models in table3

---

> ### Author Response · Authors · 2022-11-17
> **Author Response**
>
> Thank you for taking the time to read our paper and your constructive feedback! We answer your questions as follows:
>
> 1. The distribution we refer to in RQ2 is the number of relevant passages that the datasets have been annotated with. This distribution differs not only from datasets to dataset, but also from training split to evaluation split. Because we mark the candidate passages selected by the non-perfect retriever during training, with a recall<100%, therefore we can only train the model to select all the known relevant passages during training, which have been retrieved. This number, however, differs from the expectation of the evaluation split (at a recall level of 100%). This leads to the described mismatch.
> 2. In our experiments the f_k function is instantiated as the selection of the first-k encoded vectors. We kept the formulation general, as future work may investigate more elaborate methods such as windowed averages, pooling operations, etc. We wanted to keep the method as simple as possible to showcase the advantages already with a rather simple starting point.
> 3. Thank you for this pointer - We explicitly added the references (previously cited in other parts of our paper). We referred to KGI, by Glass et al (https://arxiv.org/pdf/2108.13934.pdf Table 6) and the analysis paper by Shuster et al. (https://arxiv.org/pdf/2104.07567.pdf Table 1 & 16).
> 4. KG-FiD is a very interesting and cool work and the knowledge graph infusion is highly likely to be complementary to our FiD-Light approach. While their approach to use the encoder as ranker directly and subsequently introduce a cutoff of the number of passages is a valid and commonly used technique (and complementary to our vector reduction), we believe having access to as many passages as possible during the decoding is the more effective way to go: The pairwise (pair of 1 query and 1 passage) relevance scoring of the encoder has no input of the other passages from the candidate list. Therefore, relevance scores are computed in isolation. We believe our decoder-time source pointing method has a conceptual advantage over the encoder-based ranking: Having access to all encoded passages, when making the ranking decision. Now, we are able to train the model to ignore near duplicates, make a judgment that no relevant passages are present, or assemble multiple-but-distinct partially relevant passages. Our FiD-Light model, even with T5-Base, empirically outperforms all related methods on the ranking task on 6/7 KILT tasks (see Table 6 - R-Precision columns). This capability comes especially in hand with tasks like HotpotQA, where two distinct partially relevant passages need to be positioned at the top. FiD-Light with source pointing sets a new HotpotQA SOTA with +9 points R-Precision on the leaderboard, without any specific multi-hop changes done to the architecture.
> 5. We base our latency reduction claims of FiD-Light vs FiD only on our own controlled measurements, presented in Figure 5, where all model latencies have been measured in the same environment and the exact same set of queries and candidate passages. Unfortunately, we can not make credible claims of efficiency compared to related methods (as presented in Table 3), because the KILT leaderboard focuses only on effectiveness. However, as most models use FiD we can assume their costs to be comparable to our FiD implementation (mainly changed by number of passages to input, where we use 40). As a ballpark comparison, albeit on different hardware & setup, in the PAQ paper of Lewis et al. (https://arxiv.org/pdf/2102.07033.pdf) they report FiD-Base with 500ms and FiD-Large with 2s latency per query.
>
> We hope we could answer your questions and look forward to your response. Thank you again for reviewing our paper!

---

### Author Response · Authors · 2022-11-17
**Overall Response**

Thank you so much to all reviewers for reviewing our paper on FiD-Light. We address individual questions from each reviewer in separate comments. In addition to our responses we improved the paper from the feedback. We improved the clarity at various points and added the asked for references. We updated the pdf on openreview. We hope we could sufficiently address and answer your questions and look forward to your response. Thank you again for reviewing our paper!

---

### Decision · Program_Chairs · 2023-01-20

**Decision:**

Reject

**Justification For Why Not Higher Score:**

Not novel enough

**Justification For Why Not Lower Score:**

NA

**Metareview: Summary, Strengths And Weaknesses:**

This paper proposes a lighter version of FiD by compressing the length of encoded vectors and using a decoding-time re-ranking to improve the top-ranked provenance precision. While the method shows significant boosts in efficiency, reviewers were not very convinced about the technical novel on both aspects (compression is to select simply top-k tokens and re-ranked has been done before although via an encoder).

**Summary Of Ac-Reviewer Meeting:**

didn't get all reviewers on board.